# A Single-Arm, Prospective, Exploratory Study to Preliminarily Test Effectiveness and Safety of Skin Electrical Stimulation for Leber Hereditary Optic Neuropathy

**DOI:** 10.3390/jcm9051359

**Published:** 2020-05-06

**Authors:** Takuji Kurimoto, Kaori Ueda, Sotaro Mori, Seiko Kamada, Mari Sakamoto, Yuko Yamada-Nakanishi, Wataru Matsumiya, Makoto Nakamura

**Affiliations:** Division of Ophthalmology, Department of Surgery, Kobe University Graduate School of Medicine, 7-5-2 Kusunoki-cho, Chuo-ku, Kobe 650-0017, Japan; kueda@med.kobe-u.ac.jp (K.U.); smori@med.kobe-u.ac.jp (S.M.); seiko.kondou@gmail.com (S.K.); mariwsakamoto@people.kobe-u.ac.jp (M.S.); yamaday@med.kobe-u.ac.jp (Y.Y.-N.); ytkmatsu@hotmail.com (W.M.); manakamu@med.kobe-u.ac.jp (M.N.)

**Keywords:** skin electrical stimulation, Leber hereditary optic neuropathy, mitochondrial DNA 11,778 mutation, intractable disease, retinal ganglion cell

## Abstract

Leber hereditary optic neuropathy (LHON) is an intractable disease associated with mitochondrial DNA (mtDNA) mutations. In this preliminary, single-arm, prospective, open-label exploratory trial, we investigated the effectiveness and safety of skin electrical stimulation (SES) for cases of LHON harboring the mtDNA 11,778 mutation. Of the 11 enrolled patients, 10 completed six sessions of SES once every two weeks over a 10-week period. The primary outcome measure was the change in logarithm of the minimum angle of resolution (logMAR)-converted best-corrected visual acuity (BCVA) at one week after the last session of SES. The main secondary outcome measures were the logMAR BCVA at four and eight weeks and Humphrey visual field test sensitivities at one, four, and eight weeks. At all follow-up points, the logMAR BCVA had improved significantly from baseline, [1.80 (1.70–1.80) at baseline, 1.75 (1.52–1.80) at one week, 1.75 (1.50–1.80) at four weeks, and 1.75 (1.52–1.80) at eight weeks; *p* < 0.05]. At eight weeks of follow-up, five patients showed >2-fold increase in the summed sensitivity at 52 measurement points from baseline. No adverse effects were observed. In conclusion, SES could be a viable treatment option for patients with LHON in the chronic phase harboring the mtDNA 11,778 mutation.

## 1. Introduction

Leber hereditary optic neuropathy (LHON) is a rare genetic disease in which retinal ganglion cells (RGCs) undergo cell death due to mitochondrial DNA (mtDNA) mutations [1,2]. The pathogenesis of LHON seems to involve a combination of decreased complex I driven ATP production, increased reactive oxygen species production, and, ultimately, RGC apoptosis [3]. Previously, several clinical trials have attempted to improve vision by using idebenone, coenzyme Q10 derivatives [4], EPI-473 [5], and gene therapy via an adeno-associated virus serotype 2 (AAV2) vector [6]. However, the effectiveness of these approaches remains unclear, and no firm evidence has been established in favor of any treatment for LHON.

Transcorneal electrical stimulation (TES) has shown some promise as a treatment for activating the retina and optic nerve. It has been shown to improve visual function in cases of traumatic optic neuropathy [7], non-arteritic anterior ischemic optic neuropathy [7], and retinal artery occlusion [8,9]. The neuroprotective effects of TES are thought to be attributable to the upregulation of insulin-like growth factor-1, brain-derived neurotrophic factor (BDNF), ciliary neurotrophic factor in Müller cells, and Bcl-2 homologous antagonist/killer [10,11,12,13]. However, TES has the potential to cause corneal epithelial damage because the stimulating electrode is placed directly on the cornea. Thus, repeatedly applying TES, which is necessary to maintain RGC re-activation, is a challenge in clinical settings. By contrast, skin electrical stimulation (SES), where the stimulating electrode and a reference electrode are placed on the skin around the eyes, is not only free from corneal complications but also can elicit the necessary phosphene response (as with TES). Phosphene response serves a useful index of the electrical stimulation of RGC and has been used for the evaluation of implanted artificial retinas [14,15]. This improved safety profile may allow SES to be applied repeatedly to re-activate damaged RGCs, making it more suitable than TES. In a recent report, for example, SES improved visual acuity and visual fields in patients with retinitis pigmentosa, without causing adverse effects [16].

In the present pilot study, we investigated the effectiveness and safety of SES in cases with LHON harboring the mtDNA 11,778 mutation. Our interest focused on whether the repeated application of SES could improve their visual function and whether it caused any adverse effects.

## 2. Materials and Methods

### 2.1. Study Design and Ethics

This was an exploratory, non-randomized, single-arm feasibility trial to determine the efficacy and safety of SES for patients with LHON and the mtDNA point mutation at position 11,778. The study was conducted in accordance with the Declaration of Helsinki, and the protocol was approved by the Ethics Committee of Kobe University, Japan (approval No. 180052). The detailed protocol has been published elsewhere [17].

Before being enrolled, all subjects were required to have given their informed consent for both participation and the publication of results. Specifically, we provided written and verbal information about the exact nature of the study, the implications and constraints of the protocol, the known side effects, and any risks of participating. It was clearly stated that participation was voluntary and that they were free to withdraw at any time, for any reason, without prejudice to future care, and with no obligation to give the reason for withdrawal.

### 2.2. Trial Registration

The clinical trial was registered at two locations according to the policy revision of the Japanese government: (1) the University Hospital Medical Information Network-Clinical Trial Registry: registration identifier, UMIN000031057, registered 20 March 2018 (https://upload.umin.ac.jp/cgi-open-bin/ctr/ctr_view.cgi?recptno=R000035434), according to the 2014 guideline for clinical trials by the Ministry of Education, Culture, Sports, Science and Technology and the Ministry of Health, Labor and Welfare of Japan; and (2) the Japan Registry of Clinical Trials: registration identifier, jRCTs052180066, registered 18 February 2019 (https://jrct.niph.go.jp/re/reports/detail/619), according to the Clinical Trail Act of April 2018.

### 2.3. Inclusion and Exclusion Criteria

The inclusion and exclusion criteria, together with our rationale, have been published in detail elsewhere [17]. However, we provide a summary of the criteria participants were required to meet (Table 1). Regarding the selection of the eye to be stimulated, we were led by patient preference in the first instance; if no preference was given, we selected that with the worst visual acuity.

### 2.4. Interventions

Detailed plans of the treatment session have been published elsewhere [17]. Over a 10-week period, participants received six sessions of SES, with one session every 2 weeks. To generate SES, the Mayo corporation (Aichi, Japan) provided the non-approved medical instrument used in this study, which monitored the electrical current being applied to the body and prevented excessive electrical stimulation. For the intervention, two electrode pads were placed on the skin around an eye, one on the forehead above the eyebrow and the other below the margin of the lower lid (Figure 1). If participants had any visual function, we judged that they noticed phosphene elicited by the SES procedure. The minimum current at which a participant perceived phosphene throughout the entire visual field was designated as a threshold current and was applied to that participant during SES treatment. The participant was then treated with the following SES protocol for 30 min: biphasic square wave, 1 mA amplitude, 10 ms duration, and 20 Hz frequency. In cases where the phosphene threshold exceeded 1 mA, the maximum current of 1 mA was applied for the sake of safety. The investigator monitored reactions of participants during SES.

### 2.5. Outcomes

#### 2.5.1. Parameters

The main goal of this study was to determine the efficacy and safety of SES. At baseline, we collected the following data for participants: gender, age, logMAR best-corrected visual acuity (BCVA), Humphrey visual field (HVF) sensitivity, critical flicker frequency (CFF), color vision, signal-to-noise ratio (SNR) area under the curve (AUC) for receiver operating characteristic analysis of the multifocal visual evoked potential (mfVEP), ganglion cell complex (GCC), and the circumpapillary retinal fiber layer (cpRNFL) thicknesses, central sensitivity in the macula by MP-3 microperimetry (Nidek Co., Ltd., Aichi, Japan), and corneal endothelial cell densities obtained by specular microscopy (Konan Medical, Nishinomiya, Japan). Regarding logMAR BCVA, we defined finger counting and hand motion as 1.8, light perception as 1.9, and loss of light perception as 2.0, according to the International Council of Ophthalmology standard [18]. HVF sensitivity was assessed by the 30-2 Swedish Interactive Thresholding Algorithm (SITA; Carl Zeiss Meditec, Inc., Dublin, CA, USA), using a size V visual stimulus (easier than size III for detecting subtle visual field changes in patients with LHON) [19]. Color vision was tested by standard pseudoisochromatic plates, part 2, for acquired color vision defects (SPP-2; JFC Sales Plan Co. Ltd., Tokyo, Japan). The GCC and cpRNFL thicknesses were obtained by spectral domain optical coherence tomography (OCT, Topcon Corporation, Tokyo, Japan). For OCT, data were excluded if quality scores were ≤ 30, if algorithm line failures occurred, or if poor fixation produced images with low quality.

#### 2.5.2. Primary and Secondary Outcomes

Participants underwent follow-up examinations at one, four, and eight weeks after the last session of SES. The primary outcome measure was the change in logMAR BCVA at one week. Secondary outcome measures for evaluating the effectiveness included the HVF test and the CFF at one, four, and eight weeks. However, we also assessed the following secondary measures at eight weeks: the SPP-2 results, the GCC and cpRNFL thicknesses, the number of measurements with a >5 dB increase in MP-3 sensitivity, and the SNR-AUC for the mfVEP.

#### 2.5.3. Calculating the SNR-AUC

The SNR-AUC for the mfVEP was calculated using VERIS 5.2 software (Electro-Diagnostic Imaging, Inc., Redwood City, CA, USA), according to our previous reports [20,21]. In brief, SNRs were calculated by dividing the root-mean-square amplitude of each local response by the average of 60 root-mean-square amplitudes of the noise window. The largest SNR values were selected from both windows by comparing the responses from either one of the two vertical channels or the horizontal channels. We then plotted the proportion of mfVEP responses that exceeded a specific SNR criterion for the signal window (hit rate) against the proportion that exceeded the SNR criterion for the noise window (false positive rate). The highest AUC values obtained from either pool were defined as SNR-AUCs for that eye. Probably plots were then estimated, as previously described [20], before evaluating the SNR-AUC by dividing into central 36 recording sectors corresponding to those within 10° of the HVF central 30-2 SITA program and total 60 recording sectors corresponding to those within 22° of the HVF central 30-2 SITA program.

### 2.6. Sample Size Determination and Power

We determined the sample size based on the historical demographics of our cohort. The mean and standard deviation logarithm of the minimum angle of resolution (logMAR) BCVA for 14 cases of LHON with the mutation at position 11,778 followed up at our hospital was 1.65 ± 0.49. Therefore, we considered it reasonable to conclude that SES was effective if treatments improved the logMAR BCVA to 1.0, given the otherwise low rate of spontaneous recovery of LHON patients with the 11,778 mutation [22,23]. Assuming an experimental hypothesis that the difference of the averaged logMAR BCVA from before the study to one week after the final session was 0.65 ± 0.49, we used the paired Student’s t-test with a two-sided significance level of 5% to confirm the null hypothesis of zero difference between assessments. This revealed that we needed 10 cases to achieve a study power of 80%. Allowing for possible drop-out after participation, we decided that 11 cases were optimal because further drop-out to nine would only cause the statistical power to fall to 70%.

### 2.7. Statistical Analysis

The primary endpoint and secondary visual acuity and field endpoints were analyzed in the full analysis set (i.e., all randomly assigned cases that received consecutive SES and underwent all efficacy assessments). Normality was tested by Shapiro–Wilk test. Primary and secondary outcome measures comparing data at non-repeated test points were analyzed by Wilcoxon signed-rank test. Regarding secondary endpoints for repeated measurements, in case of no rejection of normality, efficacy outcomes for repeated measures were analyzed by the mixed-effects model, using the least squares distance to address all available post-baseline data and determine if the differences from baseline to each measurement point were statistically significant. In case of rejection of normality, Wilcoxon singed-rank test was applied, in which the multiplicity for two tests is avoided using the closed test procedure such that the test of difference between four weeks and baseline was conducted only if the test of between eight weeks and baseline were rejected. Pearson’s correlation coefficients were used to evaluate associations between the difference in the summed actual sensitivity between baseline and eight weeks after the last session of SES and the following variables: age at onset, age at treatment, and intervals between disease onset and SES. Finally, the frequency of adverse events was used as the safety endpoint.

Continuous variables were expressed as means ± standard deviations for Gaussian distribution and as medians (interquartile ranges) for non-Gaussian distribution, whereas categorical variables for the descriptive data were expressed as frequencies and proportions. Unless stated otherwise, the results are presented as two-sided *p*-values, with *p* < 0.05 considered statistically significant. All analyses were performed using IBM SPSS (SPSS, Chicago, IL, USA).

## 3. Results

### 3.1. Baseline Clinical Data

We screened and enrolled 11 cases, among which one withdrew consent because the subject felt no benefit from SES. The remaining 10 cases completed all SES sessions, and their data were included for statistical analysis. We summarize their baseline demographic and clinical characteristics in Table 2. All cases were male and the average time from the onset of LHON to the initiation of SES was 154 ± 135 months. LogMAR BCVA and SNR-AUC data were not normally distributed, whereas other continuous variables were normally distributed.

### 3.2. Primary Outcome Measure

The primary outcome measure was the change in LogMAR-converted BCVA from baseline to one week after the last session of SES. The median of logMAR BCVA values at baseline and one week were 1.80 (1.70–1.80) and 1.75 (1.52–1.80), respectively, which were statistically significantly different (Wilcoxon signed-rank test; *p* = 0.042) (Figure 2).

### 3.3. Secondary Outcome Measures

The median of logMAR BCVAs at four and eight weeks after SES were 1.75 (1.50–1.80) and 1.75 (1.52–1.80), respectively. These were significantly lower (i.e., improved) from baseline (*p* = 0.043 and *p* = 0.042, respectively).

Concerning the CFF, there were no significant differences from baseline to any follow-up after SES (Figure 3a). However, four cases exhibited increased CFF from baseline to one week of follow-up: two showed increases >20%, and another two showed increases of 10–20% (Figure 3b). Over time, the cases with a CFF increase of 10–20% returned to the baseline level. Another case showed no significant changes in CFF during follow-up, and the remaining five cases actually showed decreased CFF values.

The time course of changes in summed actual sensitivity at the 52 measurement points excluding the outermost points using HVF stimulus size V was evaluated at one, four, and eight weeks after SES. The summed actual sensitivity doubled in five cases during the follow up and remained unchanged in the remaining five cases (Figure 4a,b). In other words, the summed actual sensitivity in the former significantly increased from baseline after SES (a mixed effect model, F value = 10.578, *p* = 0.01), while that in the latter did not (F value = 1.326, *p* = 0.312). Figure 4c shows the results for four representative cases that exhibited marked improvements (Cases 5, 7, and 10) and deterioration (Case 9) in their visual fields over time. In Case 9, although the logMAR BCVA remained unchanged (1.8 throughout), the summed actual sensitivity in the stimulated eye decreased in the 52-point group from 180 dB at baseline to 14 dB at the end of the follow-up. This case fulfilled all inclusion criteria at screening but had a much more recent onset of LHON than the other cases. Notably, the logMAR BCVA in the untreated eye deteriorated from −0.18 to 1.3 due to a large central scotoma one month before the study (the optic disc in that eye was still swollen at the first session of SES).

To further identify factors associated with the visual field changes, we evaluated the correlation between the difference in the summed actual sensitivity between baseline and at 18 weeks with age at onset (Figure 5a), age at treatment (Figure 5b), and interval between the disease onset and SES (Figure 5c). The change in the summed sensitivity had a significant correlation with age at onset (*r* = −0.67, *p* = 0.03), whereas it did not with age at treatment (*r* = −0.46, *p* = 0.18) or the duration between disease onset and SES (*r* = 0.06, *p* = 0.86). This suggests that the younger is the patient at onset, the more effective is SES, irrespective of the interval between disease onset and SES initiation.

Regarding the mfVEP, we used the SNR-AUC (range, 0.5–1.0) as an index analogous to the total deviation of the HVF test. An SNR-AUC value of 0.5 indicated that there was no significant signal, whereas a value of 1.0 indicated that the whole stimulated sector has significant signals. We previously showed that this corresponded well to glaucomatous visual field changes [20]. The median of SNR-AUC in 60 recording sectors was 0.54 (0.51–0.60) at baseline and 0.57 (0.56–0.62) at eight weeks after the last SES session (*p* = 0.33; Figure 6a). Similarly, the average SNR-AUC in the central 36 recording sectors was 0.57 (0.55–0.60) at baseline and 0.58 (0.54–0.61) at eight weeks (*p* = 0.54; Figure 6b). Thus, SNR-AUC in both 60 recording sectors and central 36 recording sectors were not significantly increased compared with the baseline. Table 3 lists the summary of changes in visual function parameters tested. In other words, four cases showed BCVA improvement, two cases CFF improvement, and five cases VF sensitivity improvement, whereas no case exhibited BCVA deterioration, one case VF CFF deterioration, and one case VF sensitivity deterioration. Regarding color vision and central visual sensitivity detected by MP-3 microperimetry, neither could be performed because of the severe central visual field defect (data not shown). 

### 3.4. Safety and Tolerability

All SES sessions were completed in all 10 cases. None expressed tolerability issues relating to the electrical stimulation. We also observed no local keratitis, dermatitis, anterior ocular segment inflammation, optic media opacity, fundal abnormalities, facial and trigeminal nerve abnormalities, and nasal abnormalities. There were also no issues with the electrical stimulation device during the trial. However, to evaluate the safety of SES on the retina, we measured the thicknesses of the GCC (Figure 7a) and the cpRNFL (Figure 7b). There were no significant changes in either parameter from baseline to eight weeks after the last SES session (GCC: a mixed effect model; F value = 2.30, *p* = 0.13 cpRNFL: F value = 1.46, *p* = 0.27) (Figure 7a,b). Corneal endothelial cells did not significantly reduce from baseline to eight weeks after the last session of SES (F value = 0.50, *p* = 0.69) (Figure 7c).

## 4. Discussion

### 4.1. Summary

In this preliminary study, we demonstrated that repeated application of SES led to significant improvements in both visual acuity (primary outcome) and visual fields (major secondary outcome) in 10 cases of LHON with the mtDNA 11,778 mutation that otherwise rarely shows spontaneous recovery [24]. Visual fields also continued to improve in five cases until eight weeks after the final SES session. Moreover, none of the 10 cases experienced any of the predicted SES-related adverse events or discontinued therapy because of skin sensory irritation. Our data, therefore, indicate that SES has the potential to be an effective and safe treatment for patients with LHON. To date, several other promising therapies have been used with mixed results in improving visual function. These include gene therapy and treatment with either idebenone or EPI-743, and these are discussed next.

### 4.2. Comparison with Treatment Alternatives

Regarding gene therapy, Guy et al. delivered the MT-ND4 gene via an AAV2 vector to five (legally) blind cases of LHON with the 11,778 mtDNA mutation. In two cases, the visual acuity improved over 90 days from hand motion to seven and 15 letters (equivalent to three lines of logMAR visual acuity) and without any serious adverse effects [6]. Wan et al. reported that a single-dose intravitreal injection of rAAV2-ND4 in nine cases of LHON with the 11,778 mtDNA mutation resulted in the logMAR visual acuity improving by >3 lines in six cases over a nine-month period [25]. Recently, Bouquet et al. implemented gene therapy for cases of LHON with the 11,778 mtDNA mutation [26]. They sought to confirm the safety of rAAV2/2-ND4 for 96 weeks after injection and found that 13 of their 15 cases experienced intraocular inflammation, with mild anterior chamber inflammation and vitritis found at all doses. Nevertheless, all complications responded to steroid therapy.

Idebenone stimulates ATP formation by bypassing the impaired mitochondrial complex I and giving protection as a potent free radical scavenger. Its use has been assessed in the multi-center double-blind, randomized, placebo-controlled RHODOS trial (The Rescue of Hereditary Optic Disease Outpatient Study) among 85 cases of LHON with mtDNA mutations at positions 3460, 11,778, and 14484 [4]. Unfortunately, the primary outcome of the best recovery in visual acuity after 24 weeks of therapy with 900 mg/day idebenone was not significantly different between the placebo and idebenone groups. However, the data show a positive trend for the secondary outcome, including change from baseline in best visual acuity and change in visual acuity for both eyes, particularly in cases with discordant visual acuities.

EPI-743 is a similar drug to idebenone that has been trialed for the treatment of mitochondrial diseases such as LHON and Leigh syndrome [5,27]. It is a short-chain analog of ubiquinone that targets the repletion of reduced intracellular glutathione [28]. In a small open-label trial among five consecutively treated cases of LHON with mtDNA mutations (three with 11,778, one with 14,484, and one with 3460), EPI-743 arrested disease progression and reversed vision loss in four cases. One of three cases with the 11,778 mtDNA mutation showed marked improvement and the case with the 11,448 mtDNA mutation showed complete remission after starting oral treatment with EPI-743 [5]. However, in the latter case, it was unclear if the remission occurred spontaneously. The other two cases responded well to EPI-743. These data clearly show that visual function can be improved through effects on the mitochondria.

Consistent with these results, our data also show that there was significant improvement of logMAR BCVA and the actual visual field sensitivities increased more than two-fold from baseline in five cases. Further studies with a larger sample size and a comparative design to other treatment options are warranted to show confirmative evidence of SES effectiveness in patients with LHON.

### 4.3. Prognostic Factors

Notably, our results indicate that the visual field improvement following SES negatively correlated with the age at onset, irrespective of the length from symptoms onset to SES therapy. Other studies have reported on the prognostic factors for spontaneous recovery of LHON, providing evidence for having an early age of onset, a slowly progressive course, a 14,484 mtDNA mutation, a thicker nerve fiber layer on OCT, and a larger diameter of the optic disc.

Mashima et al. reported on 61 cases of LHON harboring the 11,778 mtDNA mutation, among which 31 had an age at onset of ≤18 years and 30 had an age at onset of ≥19 years. The group that was younger at onset had a better final visual acuity than the older group [29]. Moon et al. also retrospectively investigated whether clinical and ophthalmic findings were related to the prognosis of visual acuity in LHON between the groups with and without visual acuity recovery of more than three lines of logMAR. The group with the greatest recovery had significantly younger ages at onset and fewer cases with either disc hyperemia or peripapillary telangiectasia in the fundus [30].

The recent gene therapy study that used rAAV2-ND4 for LHON cases with the 11,778 mtDNA mutation also investigated which baseline parameters were correlated with visual prognosis. At three months after injection, they found significant differences in baseline visual acuity and visual field index, but not age at onset, between the groups with and without a three-line improvement in logMAR. Logistic regression analysis confirmed that only baseline visual acuity and visual field index were significant in the visual acuity improvement [31], indicating no role for age at onset.

Although there is no clear association between baseline parameters and the effectiveness of SES, we must assess more cases to determine the factors that predict visual function improvements with this treatment. However, given the favorable outcomes in younger subjects to both SES and other treatments, irrespective of the interval from onset to treatment, there may be most hope for patients with younger onsets of LHON. It may be that the RGCs in these patients are either alive but dysfunctional or capable of being regenerated due to the high plasticity of progenitor cells.

### 4.4. The Mechanisms of SES

We are aware of no prior research into the mechanism underlying the visual improvement elicited by electrical stimulation in LHON cases. A recent study of transdermal electrical stimulation has shown visual field improvement in cases of retinitis pigmentosa [16]. TES has also been suggested to slow the progression of visual field deterioration in cases of open-angle glaucoma [32]. Other randomized prospective studies have shown no adverse events even when using more SES sessions [33,34]. In contrast to the lack of knowledge for SES, the pathological mechanisms of LHON have been thoroughly investigated. There have also been studies into electrical stimulation and the role of BDNF from which we can glean some useful information.

The principal pathological mechanism in LHON is that dysfunction of respiratory chain complex I, induced by a mitochondrial gene mutation, leads to reduced ATP production and excess reactive oxygen species production. The exacerbated oxidative stress resulting from mitochondrial gene mutation causes mitochondrial DNA damage and further reduces the levels of antioxidants such as glutathione peroxidase, glutathione reductase, superoxide, manganese superoxide dismutase, and zinc–copper superoxide dismutase. Cell death signaling also appears to be related to the mitochondrial gene mutation, which is thought to suppress the release of Bcl-2 and to activate apoptosis regulator BAX and Bcl-2 homologous antagonist/killer. These result in the release of cytochrome C, and, eventually, the cleaved active caspase executes RGC apoptosis [35].

Studies have shown that electrical stimulation of the retina increases the expressions of insulin-like growth factor-1, BDNF, and Bcl-2 in rodent retinas, enhancing RGC or photoreceptor cell survival in cases of optic nerve injury or light-induced photoreceptor degeneration [10,11]. Additionally, both in vitro and in vivo studies of electrical stimulation applied to cultured muscle cells or mouse peripheral nerves have shown enhanced mitochondrial biogenesis and anterograde axonal transport [36,37].

Regarding the relationships between neurotrophic factors and mitochondria, BDNF appears to enhance mitochondrial biogenesis. The in vitro report discussed above clearly showed that BDNF exerts protective effects on excitotoxic brain lesions induced by ibotenate, a glutamate analog, and increases the respiratory control index through the mitochondrial respiratory chain. By contrast, it showed that other neurotrophic factors, such as nerve growth factor and glial-derived neurotrophic factor, did not induce these effects. When researchers applied BDNF in combination with rotenone the effect of BDNF was inhibited in a concentration-dependent manner [38], confirming that the protective effects of BDNF on mitochondria are mediated through respiratory chain complex I.

Given the pathological mechanisms of LHON and the effects of electrical stimulation on the eye, SES is likely to upregulate BDNF and Bcl-2 expression in the retina. In doing so, it probably boosts mitochondrial biogenesis in RGCs, leading to enhance visual function.

### 4.5. Limitations

There are several limitations in the present study, not least of which are the small number of enrolled cases, the short observation period, and lack of a control arm. Ideally, a randomized trial would be best to show firm evidence of SES effectiveness in patients with LHON. Furthermore, the comparative analysis of SES effectiveness should be performed by the application of the different strength of electrical stimulation to the cases including sham stimulation (0 mA) as a control group [9,39].

It should also be noted that we enrolled patients in chronic phases of LHON with long intervals between onset and treatment; as such, we do not know the therapeutic effectiveness of SES on acute LHON. However, in the one case with a recent onset, we actually found that their visual field deteriorated after starting SES. Although this case fulfilled the inclusion criteria, more than eight months passed without visual improvement after the onset of the eye to be stimulated. The deterioration of visual function might be natural disease progression accompanied with the onset of untreated fellow eye. From another perspective, given this marked difference in response compared with the other cases, cell signaling activated by electrical stimulation in RGCs or glial cells may diverge between acute and chronic phases of LHON, and applying SES may be unsuitable in the acute phase. 

Moreover, animal experiments have shown that repeated electrostimulation has a greater pro-survival effect than single electrostimulation of RGC after optic nerve damage [40,41]. Further studies are warranted to elucidate whether more frequent SES has more and sustained impact on visual improvement compared to stimulation every two weeks as conducted in the present study. Finally, we did not objectively evaluate the threshold for eliciting phosphene. This means we lack clarity about the correlation between the degree of electrical stimulation that functionally activated the retina and the degree of improvement in visual function.

## 5. Conclusions

This preliminary study showed that six sessions of SES, one every two weeks, induced significant improvements to both visual acuity and visual fields among patients with LHON in the chronic phase harboring the 11,778 mtDNA mutation. Furthermore, visual field recovery after SES was significantly associated with a younger age at onset. To establish SES as a novel therapy for LHON, it will be necessary to elucidate the mechanism underpinning its neuroprotective effects and to examine the long-term effectiveness on LHON after frequent applications, ideally by a placebo control, randomized clinical trial.

## Figures and Tables

**Figure 1 jcm-09-01359-f001:**
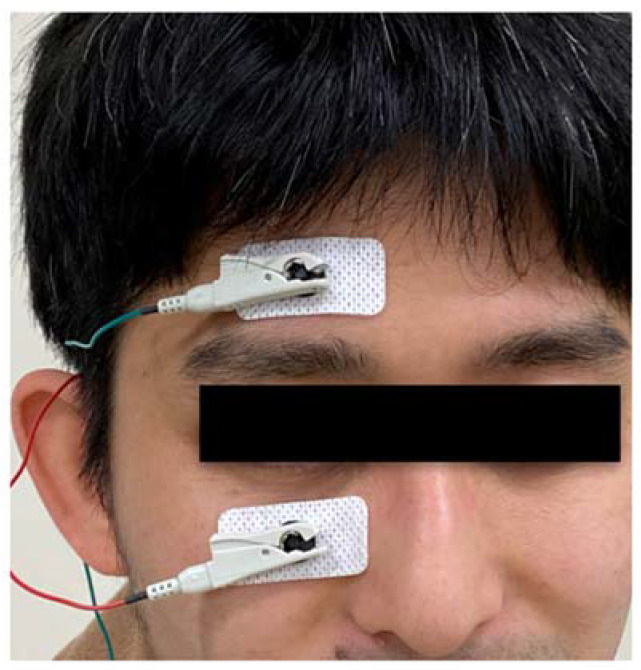
The location of the electrical pads.

**Figure 2 jcm-09-01359-f002:**
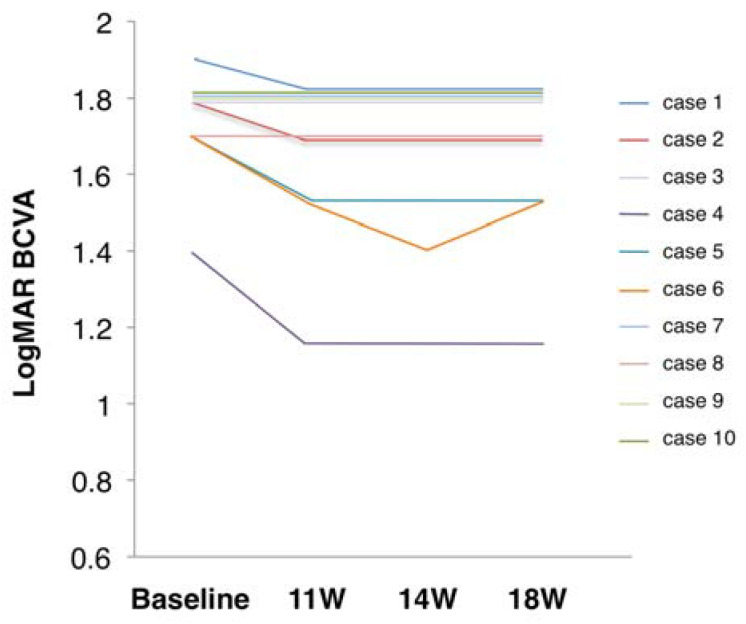
Changes in visual acuity from before to after skin electrical stimulation. LogMAR BCVA, logarithm of minimal angle resolution-converted best-corrected decimal visual acuity. Courses of individual cases.

**Figure 3 jcm-09-01359-f003:**
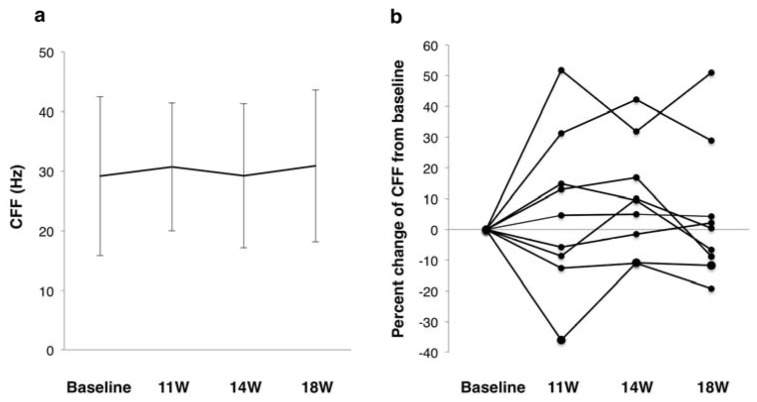
Changes in CFF before and after SES: (**a**) mean changes, where error bars indicate standard deviations; and (**b**) percent change from baseline. CFF, critical flicker frequency; SES, skin electrical stimulation.

**Figure 4 jcm-09-01359-f004:**
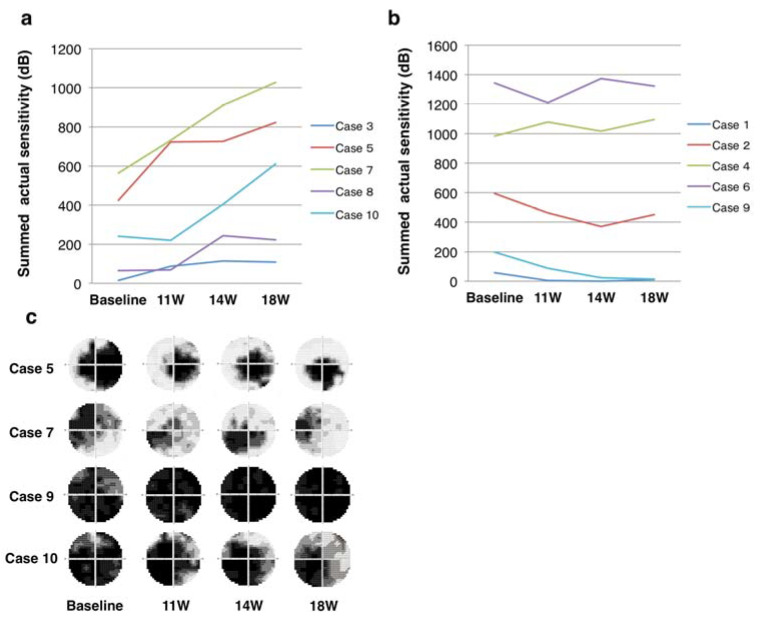
Time course of changes in summed actual sensitivity at 52 test points (except outermost points). (**a**) the effective group showing the increase of summed actual sensitivity compared with baseline; and (**b**) ineffective group showing no improvement. (**c**) The Humphrey visual field grayscale in four representative cases that had improved visual fields (Cases 5, 7, and 10) and deterioration of visual field (Case 9).

**Figure 5 jcm-09-01359-f005:**
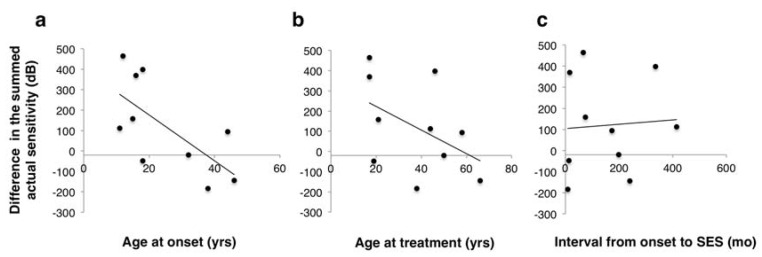
Scatter plots of the difference in summed actual sensitivity between baseline and 18 weeks as a function of: (**a**) age at onset (*r* = −0.67, *p* = 0.03); (**b**) age at treatment (*r* = −0.46, *p* = 0.18); and (**c**) the interval between disease onset and skin electrical stimulation (*r* = 0.06, *p* = 0.86).

**Figure 6 jcm-09-01359-f006:**
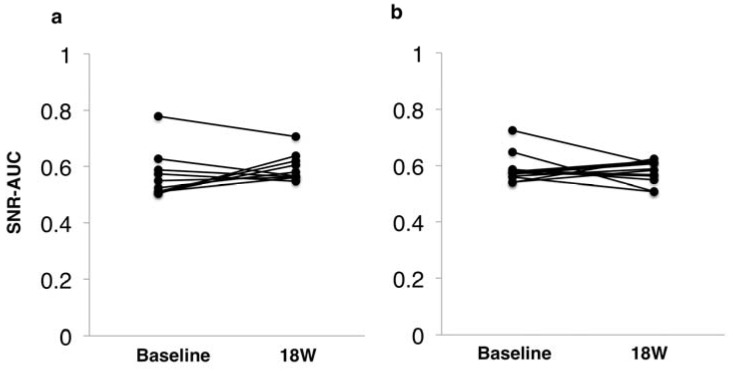
ROC analysis for multifocal visual evoked potentials from baseline to eight weeks after SES: (**a**) all 60 sectors showed no significant difference in SNR-AUC (*p* = 0.33); and (**b**) the central 36 sectors showed no significant difference in SNR-AUC (*p* = 0.54). ROC, receiver operating characteristic curve; SES, skin electrical stimulation; SNR-AUC, signal-to-noise ratio of area under the curve.

**Figure 7 jcm-09-01359-f007:**
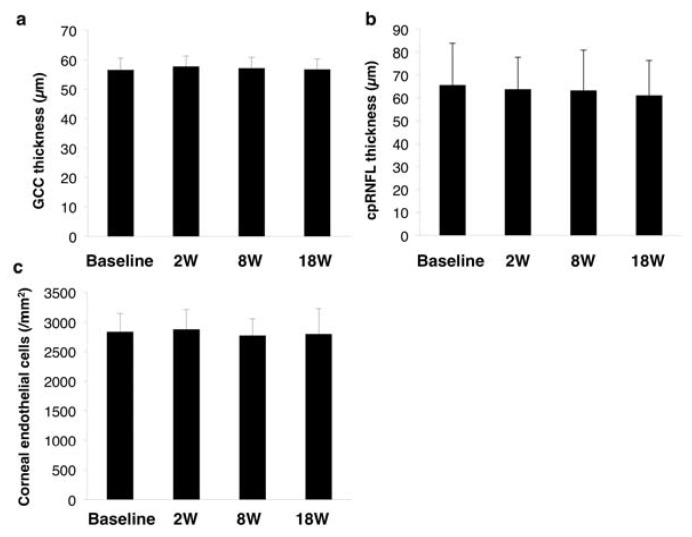
Changes in safety parameters before and after SES: (**a**) the GCC thickness; (**b**) the cpRNFL thickness measured by 3D-OCT; and (**c**) corneal endothelial density measured by specular microscopy. cpRNFL, circumpapillary retinal nerve fiber layer; GCC, ganglion cell complex; OCT, optical coherence tomography; SES, skin electrical stimulation.

**Table 1 jcm-09-01359-t001:** Inclusion and exclusion criteria.

**Inclusion Criteria**
Aged ≥ 16 and < 80 years (males and females)
More than 8 months without visual improvement
Mutation of mt DNA at the position 11,778
Best-corrected decimal visual acuity <0.1
**Exclusion Criteria**
A history of smoking within past 6 months
Implanted electronic device such as cardiac pacemaker etc.
Intraocular surgery within the past 1 year
A history of eye diseases other than early-stage cataract and artificial intraocular lens
Idebenone treatment within the last year
Participants on any of the following drugs: ethambutol, chloramphenicol, linezolid, erythromycin, streptomycin, antiretroviral drugs, amiodarone, infliximab, clioquinol, dapsone, quinine, pheniprazine, suramin sodium, or isoniazid
A history of epilepsy
Current pregnancy
Severe allergic diseases, including atopic dermatitis
Currently participating in other clinical studies
Participants judged inappropriate for other reasons by responsible doctors

**Table 2 jcm-09-01359-t002:** Demographic and clinical characteristics of participants at baseline.

Parameters	Baseline
Number of cases/eyes	10/10
Age at enrollment (years)	37.6 ± 17.2
Male/Female	10/0
Age at onset (years)	25.0 ± 12.9
Intervals from onset to SES (months)	154 ± 135
LogMAR BCVA	1.80 (1.70–1.80)
Critical flicker frequency (Hz)	29.1 ± 13.3
Ganglion cell complex thickness (μm)	56.6 ± 4.0
cpRNFL thickness (μm)	65.6 ± 18.1
Corneal endothelial cell density (/mm^2^)	2855 ± 272
SNR-AUC in all 60 sectors	0.54 (0.51–0.60)
SNR-AUC in central 36 sectors	0.57 (0.55–0.60)

cpRNFL, circumpapillary retinal nerve fiber layer; LogMAR BCVA, logarithm of minimal angle resolution-converted best-corrected decimal visual acuity; SNR-AUC, signal-to-noise ratio of area under the curve. LogMAR and SNR-AUC are expressed as median (interquartile ranges), whereas other continuous variables are expressed as mean ± standard deviation.

**Table 3 jcm-09-01359-t003:** The summary of the changes after SES in visual acuity, CFF, and visual field in each case.

Case	LogMAR BCVA	CFF	Summed Actual Sensitivity (dB)
1	1.90 → 1.80	NR	↓↓
2	1.80 → 1.70	↑	→
3	1.80 → 1.80	↓	↑↑
4	1.40 → 1.15	→	→
5	1.70 → 1.52	→	↑
6	1.70 → 1.40	↑	→
7	1.80 → 1.80	→	↑
8	1.70 → 1.70	→	↑↑
9	1.80 → 1.80	→	↓↓
10	1.80 → 1.80	→	↑↑

LogMAR best-corrected visual acuity (BCVA); baseline vs. the lowest logMAR BCVA during observation; CFF: ↑ ≥20% increase, ↓ ≤20% decrease, at 11 weeks, and NR, no response; summed actual sensitivity: ↑ or ↓ ≥50% increase or decrease and ↑↑ or ↓↓ ≥100% increase or ≥80% decrease at 18 weeks; CFF, critical flicker frequency.

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
