# Peer review of "A Single-Arm, Prospective, Exploratory Study to Preliminarily Test Effectiveness and Safety of Skin Electrical Stimulation for Leber Hereditary Optic Neuropathy"

_jcm, 2020, doi:10.3390/jcm9051359_

Round 1
Reviewer 1 Report
General comments:
This study discusses the outcome of skin electrical stimulation for treatment of chronic LHON in 10 patients. There currently exists no proven effective therapy for LHON, and it would be a great achievement to find a treatment with such low risk as SES. This study does confirm that SES appears to be well tolerated with no significant adverse effects. However, the study is not adequately designed or powered to determine the efficacy of this treatment. The effects are small; mean logMAR BCVA improved from 2.00 to 1.85, which is still well below 20/800, the limit of most Snellen acuity charts. It is unclear whether this difference has any clinical significance in terms of visual functioning. The analysis of visual fields was convoluted, and standard measures of visual fields were not included. However, the major limitation of this study is the lack of a control arm, resulting in two major issues. First, that a placebo effect cannot be excluded. Second, there is a well-recognized learning effect, whereby subjects performance on acuity charts and visual fields improve with repetition. Neither of these limitations are acknowledged or addressed in the discussion of the paper. Considering the low risk profile of the treatment, there is no justifiable reason why a control group was not included. I would strongly recommend this be undertaken to allow the results of the study to be properly assessed.
Specific comments:
Line 14: In the text, you indicated that only 10 patients completed all sessions. Please be consistent.
Line 19: Include the logMAR values at baseline and follow up to give the reader an idea of the magnitude of effect
Line 31: “gene therapy via an adeno…”
Line 34-36: The statement that TES is a “proven treatment” is too strong. The trials referenced show minimal or no improvement in visual function. For example, ref 8 regarding TES for RP showed no significant benefit. Ref 10, the only controlled trial, investigated TES for RAO and showed no effect in any function except in a wave slopes, and it is not clear how this translates into vision. The other trials were small, uncontrolled and demonstrated only questionable improvement in visual function.
Line 43-44: Please explain the importance and physiologic relevance of phosphene response
Table 1, exclusion criterion #4: Please clarify the meaning of “artificial intraocular”. For all other inclusion and exclusion criteria, the language could be significantly improved (e.g. 1. History of smoking within past 6 months, 8. Current pregnancy etc)
Figure 1: this figure does not add much, and could be replaced by a line in Table 2 for “Age at enrollment”
Line 167: suggest “after the last SES session (e.g. 11 weeks after baseline)
Figure 2: Suggest including the curves for each individual subject in addition to the mean
Line 192: Suggest including the fields for the patient who worsened significantly
Line 198-207: Is there a precedent for this type of analysis? Was this a planned analysis, or only done post-hoc? Does this data manipulation have any bearing on clinical, subjective improvement? Why not also report more traditional measures of visual field defects such as mean deviation, visual field index, and foveal threshold?
Figure 4a-c: the size of the error bars in these figures is very large. The sample size calculation was only directed at detecting a change in logMAR acuity, and was clearly not large enough for this extensive manipulation of visual field data.
Line 223-226: Please explicitly state that no significant improvement was detected
Line 241: Please state the there was a brief increase in GCC thickness at 2 weeks, but return to baseline at all subsequent follow ups.
Figure 7: As there was no significant change in any of these variables, and the data is described in the text, this figure does not add anything
Line 269: Remove the word “again”, please complete the sentence
Line 291-292: This sentence is incomplete
Line 293-294: Please include the caveat that no definitive treatment has yet been discovered, so comparison of SES with these treatments may likewise only be as good as placebo.
Line 338: capitalize BDNF
Line 355: The greatest limitation of this study is the lack of a control arm
Line 358: If SES boosts BDNF, which has been shown to enhance cell survival, it would not make sense that an acute LHON would have less treatment effect. Consider the possibility that there was no effect from SES and the patient continued on their natural course.
Author Response
DATE 4/7/2020
Professor Dr. Emmanuel Andrès
Professor Dr. Michael G. Hennerici
Editor-in-chief
Journal of Clinical Medicine
Re: Kurimoto T et al. “Effectiveness and Safety of Skin Electrical Stimulation for Leber Hereditary Optic Neuropathy”
Dear editors:
We are grateful for the positive comments of the reviewers regarding our manuscript and agree with most of their criticism. As detailed below, we have revised the manuscript to confer with these comments. Substantial changes in the manuscript are indicated in red.
Reviewer 1 evaluation
- This study discusses the outcome of skin electrical stimulation for treatment of chronic LHON in 10 patients. There currently exists no proven effective therapy for LHON, and it would be a great achievement to find a treatment with such low risk as SES. This study does confirm that SES appears to be well tolerated with no significant adverse effects. However, the study is not adequately designed or powered to determine the efficacy of this treatment. The effects are small; mean logMAR BCVA improved from 2.00 to 1.85, which is still well below 20/800, the limit of most Snellen acuity charts. It is unclear whether this difference has any clinical significance in terms of visual functioning. The analysis of visual fields was convoluted, and standard measures of visual fields were not included. However, the major limitation of this study is the lack of a control arm, resulting in two major issues. First, that a placebo effect cannot be excluded. Second, there is a well-recognized learning effect, whereby subjects performance on acuity charts and visual fields improve with repetition. Neither of these limitations are acknowledged or addressed in the discussion of the paper. Considering the low risk profile of the treatment, there is no justifiable reason why a control group was not included. I would strongly recommend this be undertaken to allow the results of the study to be properly assessed.
Response: Thank you for your invaluable comment. As described in “Sample size” section of Methods, we set a sample size a priori in a reference to a historical cohort that comprised 14 LHON cases with the mtDNA 11778 mutation and who visited our clinic. From a statistical point of view, the present sample size was set with a statistical power of 80% and type I error of 5%, although the sample size is as small as 10.
We agree that the magnitude of visual acuity improvement is small, but we believe that the improvement also has a clinical significance to some degree as mentioned below.
When writing this response, we noticed that in the original manuscript of this study, we carelessly used different sets of logMAR-converted values corresponding to visual acuity less than 0.02 of decimal scale or 20/1000 of Snellen charts (i.e.; counting fingers, hand motion, light perception, and no light perception) from those used in calculation of the historical cohort. There are two ways of the conversion of these low visual acuities to logMAR values. One is the International Council of Ophthalmology (ICO) standard (Appendix 3, Visual standard. Aspects and Ranges of Vison Loss with emphasis on Population Surveys. https://www.researchgate.net/publication/248343777_Visual_Standards_aspects_and_ranges_of_vision_loss_with_emphasis_on_population_surveys)
According to this standard, visual acuities less than 0.02 of decimal scale or 20/1000 of Snellen charts other than no light perception are all converted to logMAR 1.8 or 1.9 and no light perception is converted to logMAR 2.0. We used this standard when we applied conversion of visual acuity data of the historical cohort to log MAR values.
The other method of visual acuity conversion to logMAR scale is based on the guideline by Heuer et al (Heuer DK, Barton K, Grehn F, et al. Consensus on definitions of success. In: Shaarawy TM, Sherwood MB, Grehn F eds. Guidelines on design and reporting of surgical trials. Amsterdam, Kugler; 2008.15–24、original manuscript ref 19). According to this guideline, counting fingers, hand motion, light perception, and no light perception are converted to logMAR 2.0, 2.9, 3.2, and 3.5, respectively.
We have realized that we carelessly used this guideline protocol when converting visual acuity data to logMAR values in the present study, resulting in the skewedly large average logMAR values in the original version of the manuscript.
Re-calculation of logMAR-converted visual acuity in the present cohort according to the ICO standard, which was applied to the conversion of the historical cohort data, yielded mean ± logMAR of 1.73 ± 0.12 at baseline, 1.66 ± 0.20 at 11weeks (the primary endpoint), 1.65 ± 0.21 at 14 weeks, 1.66 ± 0.20 at 18weeks (major secondary endpoints).
The 0.1 change of logMAR is equivalent to a 5-letter change of the ETDRS chart and also means that the Snellen chart visual acuity less than 20/1000 at baseline improved to the visual acuity between 20/800 and 20/1000.
Based on the decimal visual acuity scale, this change also indicates that patients who could not discriminate the Landolt ring chart at all at baseline obtained visual acuity of at least 0.02 following SES. Although this change is still indeed small, the shift from the condition where patients could not discern any letters to the letter-discernible condition is clinically meaningful, at least to some degree, in LHON patients with the mtDNA 11778 mutation, who are known to have a very low rate of spontaneous visual recovery.
In addition, it is not surprising that the logMAR visual acuity in the present cohort is still larger than in the historical cohort (=1.65) because the historical cohort included subjects who had a decimal visual acuity of more than 0.1, whereas the inclusion criterion in the present cohort was a decimal visual acuity less than 0.1 at baseline.
We have made revision by corresponding to the above discussed issues.
Speaking of visual field evaluation, we already know and have reported that it is difficult and less useful to measure the Humphrey visual field using an ordinal visual stimulus size III in patients with LHON who exhibit a large and deep central scotoma and that measurement using the stimulus size V is more useful for the assessment of visual function in patients with LHON (original manuscript ref. No.20).
Therefore, we planned to measure the Humphrey visual field in the present cohort using stimulus size V, instead of size III, a priori at the initial stage of conceptualization of study protocol, but not post hoc (See Methods, Statistical analysis in ref. 18). Given that the Humphrey visual field test measurement using size V stimuli do not calculate MD or PSD unlike the measurement using size III stimuli, it is impossible to obtain such global indices and, thus, we used summed actual sensitivities instead.
However, because visual field changes are secondary endpoints and the original analysis of visual field changes is convoluted as you pointed out, we have revised the visual field analysis with a simpler methodology as described later.
We agree with your criticism that this study lacks a control arm, which we realized is a limitation of the present study since the conceptualization of the study protocol. That is why we entitled our published article regarding the study protocol as “a single-arm, open-label, non-randomized prospective exploratory study.”
In order to emphasize that the present study is not a confirmative but exploratory and pilot study, we have revised the title of the present study to “A single-arm, prospective, exploratory study to test effectiveness and safety of skin electrical stimulation for Leber hereditary optic neuropathy”
Although it is single-arm, the present study has set a historical cohort as a control. We think it acceptable to use such a study design for an exploratory study of a rare intractable disease.
We agree concerning a possible placebo effect for the visual improvement of the present cohort. Electrostimulation makes most subjects with some residual visual function notice phosphene and subjects who receive a sham-operation would realize that they did not receive SES because they did not notice phosphene. Therefore, it is practically impossible to make two arms with one with SES and the other with sham-operation in a masked fashion. However, we have referred to the possible placebo effect in the Discussion.
A learning effect of visual field testing may have affected the results as you pointed out. However, in the present study, the stimulus size V was used instead of size III as mentioned. To our knowledge, there has been no documented study to test whether the learning effect exists when the visual field was measured using the stimulus size V. We evaluated and plotted the changes in summed visual sensitivity at 52 test points in 3 left eyes among 5 eyes of 3 subjects from the historical cohort, who underwent the test using the stimulus size V at least four times. We chose these 3 left eyes to exclude the multiplicity and dependency of both eyes from the same individuals and because the remaining subject completed the tests on only the left eye. In addition, we also calculated and plotted summed visual sensitivity repeatedly measured in the present cohort (see below Figures 1–3).
As a result, the summed actual sensitivity of 3 eyes in the historical cohort did not change or even decreased during 4 repeated measurements (Figure 1, mixed effect model, F value = 0.954, p = 0.472). There was no change in 5 non-responders to SES, either (Figure 2, F value = 1.326, P = 0.312), whereas there was a significant increase in the number of the summed actual sensitivity in the remaining 5 responders to SES in the present cohort (Figure 3, F value = 10.578, P = 0.01). Though the number is small, no changes in the 4 repeated measurements of the historical cohort might suggest that the presence of 50% cases that doubled the summed actual sensitivity in the HVF test using size V stimuli in the present cohort cannot be accounted for by the simple learning effect. However, because the visual field analysis is a secondary endpoint as you pointed out, in the revised manuscript, we have deleted the convoluted analysis, and instead we have shown the time course of changes in the summed actual sensitivity at 52 test points in 5 eyes that experienced at least double increase in that number after SES and in the remaining 5 eyes that did not.
Figure 1. Historical cohort data of LHON cases harboring the mtDNA 11778 mutation
Figure 2. Ineffective group of LHON cases following SES in the present cohort
Figure 3. Effective group of LHON cases after SES in the present cohort
Specific comments:
- Line 14: In the text, you indicated that only 10 patients completed all sessions. Please be consistent.
Response: According to your suggestion, we have revised the expression (line 17).
- Line 19: Include the logMAR values at baseline and follow up to give the reader an idea of the magnitude of effect.
Response: According to your suggestion, we have added logMAR data at baseline (lines 23-24).
- Line 31: “gene therapy via an adeno…”
Response: We apologize for the typographical error. We have deleted “with” (line 40).
- Line 34-36: The statement that TES is a “proven treatment” is too strong. The trials referenced show minimal or no improvement in visual function. For example, ref 8 regarding TES for RP showed no significant benefit. Ref 10, the only controlled trial, investigated TES for RAO and showed no effect in any function except in a wave slopes, and it is not clear how this translates into vision. The other trials were small, uncontrolled and demonstrated only questionable improvement in visual function.
Response: We agree that TES is not a proven treatment. According to your indication, we have revised the expression (line 40).
- Line 43-44: Please explain the importance and physiologic relevance of phosphene response
Response: According to reviewer's suggestion, we have added explanation about the importance and relevance of phosphene perception (lines 51–53). The phosphene response provides a useful index of electrical stimulation of RGC and has been used for the evaluation of implanted artificial retinas (references 15 and 16).
- Table 1, exclusion criterion #4: Please clarify the meaning of “artificial intraocular”. For all other inclusion and exclusion criteria, the language could be significantly improved (e.g. 1. History of smoking within past 6 months, 8. Current pregnancy etc)
Response: We apologize for these mistakes. According to your suggestion, we have carefully corrected Table 1 (revised Table 1).
- Figure 1: this figure does not add much, and could be replaced by a line in Table 2 for “Age at enrollment”
Response: As you pointed out, the information of Figure 1 and Table 2 is overlapped. We have deleted Figure 1 and renumbered the remaining Figures and revised the text accordingly.
- Line 167: suggest “after the last SES session (e.g. 11 weeks after baseline).
Response: We apologize for this typo. We have changed the phrase from “after the last SES session” to “after the last session of SES” (line 189).
- Figure 2: Suggest including the curves for each individual subject in addition to the mean.
Response: According to your suggestion, we have added individual curves (revised Figure 2a).
- Line 192: Suggest including the fields for the patient who worsened significantly
Response: According to your suggestion, we have added a series of gray scales of the patient whose visual field was worsened into revised Figure 4c.
- Line 198-207: Is there a precedent for this type of analysis? Was this a planned analysis, or only done post-hoc? Does this data manipulation have any bearing on clinical, subjective improvement? Why not also report more traditional measures of visual field defects such as mean deviation, visual field index, and foveal threshold?
As described in the response to your major concern above, we already knew and reported that it is difficult and less useful to measure the Humphrey visual field using an ordinal visual stimulus size III in patients with LHON who manifest large and deep central scotoma and that measurement using the stimulus size V is more useful for assessing visual function in patients with LHON (original and revised manuscript ref. No.20). Therefore, we planned to measure the Humphrey visual field in the present cohort using stimulus size V, instead of size III, a priori at the initial stage of conceptualization of study protocol, but not post hoc (See Methods, Statistical analysis in revised manuscript ref. 18). Given that the Humphrey visual field test measurement using size V stimuli do not calculate MD or PSD unlike the measurement using size III stimuli, it is impossible to obtain such global indices and, thus, we used summed actual sensitivities instead
- Figure 4a-c: the size of the error bars in these figures is very large. The sample size calculation was only directed at detecting a change in logMAR acuity, and was clearly not large enough for this extensive manipulation of visual field data.
Response: As you pointed out, the sample size was calculated to test the significance of difference between logMAR at baseline and that at 1 week after the full session of SES as a primary endpoint. If the primary endpoint turned out to be not significant, to test the significance of the secondary endpoints would be meaningless. Fortunately, however, the primary endpoint was statistically significant and further logMAR at 4 and 8 weeks after SES (the secondary endpoints located at the higher hierarchy) was significantly better than that at the baseline. Therefore, further statistical analysis of visual field changes in the lower hierarchy is thought to be meaningful. However, given that the present study was an exploratory, but not confirmative, the present results only suggest, but do not confirm, the potential effectiveness of SES to patients with LHON and the mtDNA 11778 mutation. We have emphasized this issue in the Discussion.
- Line 223-226: Please explicitly state that no significant improvement was detected.
Response: According to your suggestion, we have added a comment that mfVEP response was not significantly improved (lines 252–253).
- Line 241: Please state the there was a brief increase in GCC thickness at 2 weeks, but return to baseline at all subsequent follow ups.
Response: Regarding the significance of the GCC increase at 2 weeks, it reached statistical significance only by the multiple comparison. However, the mixed effect model did not show a statistical significance as a whole. Therefore, the seemingly increased GCC thickness at 2 weeks is statistically meaningless. We have deleted the symbol from the Figure. We apologize for our misinterpretation of statistical evaluation on this issue.
- Figure 7: As there was no significant change in any of these variables, and the data is described in the text, this figure does not add anything.
As you pointed out, there are no significant changes of any parameters in this figure. However, because this figure depicts parameters of safety endpoints, it is of importance to demonstrate no significant changes in these parameters, indicating that SES does have a safety profile. For this reason, we decide to keep this figure as it is, while the description on the figure legend has been moved and summarized to the Results section in the revised manuscript.
- Line 269: Remove the word “again”, please complete the sentence
Response: We have deleted this sentence because the expression is redundant and has no important message.
- Line 291-292: This sentence is incomplete
Response: We have corrected this sentence (lines 328-330)
- Line 293-294: Please include the caveat that no definitive treatment has yet been discovered, so comparison of SES with these treatments may likewise only be as good as placebo.
Response: According to your suggestion, we have revised the expression (lines 330–332).
- Line 338: capitalize BDNF
Response: We have capitalized the word as BDNF (line 382).
- Line 355: The greatest limitation of this study is the lack of a control arm
Response: We agree that one of the limitations of this study is a single-arm design and thus have emphasized this point as well as revised the title of this manuscript. On the other hand, it is practically impossible to set a sham-operation group as mentioned earlier. We have also commented on it.
- Line 358: If SES boosts BDNF, which has been shown to enhance cell survival, it would not make sense that an acute LHON would have less treatment effect. Consider the possibility that there was no effect from SES and the patient continued on their natural course.
Response: There is no clear reason why SES was ineffective or even harmful in patients with LHON in the acute phase. Although it is known that electrical stimulation activates BDNF and IGF-1 signaling in the retina, no study has been reported that comprehensively investigates other signal cascades that may be activated or inactivated in the retina by SES. It is not unnatural to think that intraretinal signal pathways activated or inactivated by SES may differ between the acute and chronic stages in LHON. We do not deny the possibility that SES actually has no beneficial effect and that we have observed the natural course of spontaneous recovery as you pointed out. We have added this comment in the Discussion.
However, if SES has no effect at all, it is hard to explain that SES did even decrease visual function in some cases with LHON during the acute phase, suggesting some potential effect of SES on the retina. Moreover, animal experiments have shown that repeated electrostimulation has more pro-survival effect than single stimulation of RGC after optic nerve damage (Morimoto T et al., Optimal parameters of transcorneal electrical stimulation (TES) to be neuroprotective of axotomized RGCs in adult rats. Exp Eye Res; 2010; 90; 285-91, Tagami Y et al., Axonal regeneration induced by repetitive electrical stimulation of crushed optic nerve in adult rats. Jpn J Ophthalmol; 2009; 53; 257-266). Further studies are warranted to elucidate whether or not more frequent SES has more and sustained impact on visual improvement compared to once-per-fortnight stimulation as conducted in the present study. We have added these discussions in the revised text (lines 413–417).

Reviewer 2 Report
The present study is of great clinical interest as good therapeutic options for LHON are still lacking. Other studies have already shown a positive effect of electrostimulation on optic nerve function, although electrostimulation was also performed directly on the eye. In the present skin electrostimulation, the electrostimulation is performed in the eye environment. On the one hand, the method should be better presented, also with an illustration. On the other hand, the electric field should also be described in more detail to get a better idea of the field characteristics of the method. As the study was conducted on only 11 patients, it should be called a pilot study. The average age of the patients was quite young. It should be discussed that the response of the optic nerve may also be age-dependent. Furthermore, it would be interesting to list which specific visual phenomena the patients had during electrostimulation. According to the protocol, the sense of color was also tested. The results should also be listed.
Author Response
DATE 4/7/2020
Professor Dr. Emmanuel Andrès
Professor Dr. Michael G. Hennerici
Editor-in-chief
Journal of Clinical Medicine
Re: Kurimoto T et al. “Effectiveness and Safety of Skin Electrical Stimulation for Leber Hereditary Optic Neuropathy”
Dear editors:
We are grateful for the positive comments of the reviewers regarding our manuscript and agree with most of their criticism. As detailed below, we have revised the manuscript to confer with these comments. Substantial changes in the manuscript are indicated in red.
Reviewer 2 evaluation
The present study is of great clinical interest as good therapeutic options for LHON are still lacking. Other studies have already shown a positive effect of electrostimulation on optic nerve function, although electrostimulation was also performed directly on the eye. In the present skin electrostimulation, the electrostimulation is performed in the eye environment. On the one hand, the method should be better presented, also with an illustration. On the other hand, the electric field should also be described in more detail to get a better idea of the field characteristics of the method. As the study was conducted on only 11 patients, it should be called a pilot study. The average age of the patients was quite young. It should be discussed that the response of the optic nerve may also be age-dependent. Furthermore, it would be interesting to list which specific visual phenomena the patients had during electrostimulation. According to the protocol, the sense of color was also tested. The results should also be listed.
Response: Thank you very much for your invaluable comments. We have carefully revised our manuscript according to your suggestions as follows:
We have added a photograph to show how to place stimulation pads as shown in the new Figure 1. Regarding the electrical field, we adjusted the magnitude of electrical currents based on the phosphene perception by participants. Because we did not measure the electrical field, we do not know the distribution of the electrical field during stimulation. However, we have confirmed no electrical currents were evoked during stimulation of the skin around the opposite eye.
We agree with your criticism that this study is not a confirmative but exploratory and pilot study, and we have revised the title of the present study to “A single-arm, prospective, exploratory study to test effectiveness and safety of skin electrical stimulation for Leber hereditary optic neuropathy.”
As you pointed out, age at onset was associated with visual improvement, which had been already described in the Discussion and shown in the original Figure 7 (revised Figure 5).
In line with your suggestion, we have added a new Table 3, which lists the changes in visual functional parameters of each participant. However, we only conducted color plate tests regarding changes in color sense, so we did not assess how individual participants experienced changes in color sense during and after SES.
Sincerely,
Takuji Kurimoto
Division of Ophthalmology, Department of Surgery,
Kobe University, Graduate School of Medicine
7-5-2 Kusunoki-cho, Chuo-ku, Kobe 650-0017, Japan
Phone No: +81-78-381-6048
Fax No: +81-78-382-6059
Email Address: kuri1201@med.kobe-u.ac.jp

Round 2
Reviewer 1 Report
The authors have done a good job of addressing the cosmetic issues with the manuscript and have significantly improved the visual field analysis, but have not adequately addressed the major issue of placebo effect. I would respectfully disagree that the lack of phosphenes makes it practically impossible to include a placebo group. The very act of coming into a clinic, having electrodes placed on their face and having repeated testing with the associated personal attention would suffice to generate a placebo effect. Patients who have not previously undergone true SES would not know that they are supposed to see phosphenes. Using a historical cohort is not the same as a true control group, making this more of a case series and not a prospective trial. Considering the only questionable benefit of TES in prior studies, only a strong study with a control arm would elevate this to a viable treatment. Reconsider publication only with addition of an adequate control group, or submission to a lower impact journal.
Minor comments:
Line 43: as mentioned previously, TES resulted in no improvement in visual function in patients with RP, and did not improve visual function, only a wave slopes in RAO.
Figure 2: The individual traces will need to be slightly offset, as several traces are not visible at all
Line 229: Please correct the numbering of cases with improved visual fields (case 5, 7, and 10)
Line 402-406: As stated above, I do not agree that a sham procedure is impossible. More cases and longer follow up do not eliminate a placebo effect. You can state that you, as a group of authors, did not wish to include a placebo arm, but should not state that it is not possible.
Line 412: Please state that the deterioration in vision in case 9 could have been due simply to natural disease progression.
Author Response
4/18/2020
Professor Dr. Emmanuel Andrès
Professor Dr. Michael G. Hennerici
Editors-in-chief
Journal of Clinical Medicine
Re: Kurimoto T et al. “Effectiveness and Safety of Skin Electrical Stimulation for Leber Hereditary Optic Neuropathy”
Dear editors:
We are grateful for the positive comments of the reviewers regarding our manuscript and agree with most of their criticism. As detailed below, we have revised the manuscript to confer with these comments. Substantial changes in the revised manuscript are indicated in red.
Reviewer 1 evaluation
The authors have done a good job of addressing the cosmetic issues with the manuscript and have significantly improved the visual field analysis, but have not adequately addressed the major issue of placebo effect. I would respectfully disagree that the lack of phosphenes makes it practically impossible to include a placebo group. The very act of coming into a clinic, having electrodes placed on their face and having repeated testing with the associated personal attention would suffice to generate a placebo effect. Patients who have not previously undergone true SES would not know that they are supposed to see phosphenes. Using a historical cohort is not the same as a true control group, making this more of a case series and not a prospective trial. Considering the only questionable benefit of TES in prior studies, only a strong study with a control arm would elevate this to a viable treatment. Reconsider publication only with addition of an adequate control group, or submission to a lower impact journal.
Response: Thank you again for sparing your precious time to review our manuscript. We deeply appreciate the insightful comments provided by you.
First of all, we do not intend to deny the possibility of a placebo effect and sincerely agree to your criticism that an RCT with a placebo control group is mandatory as a confirmative prospective study to make a rigorous conclusion that SES is an effective treatment for LHON.
However, before doing so, an exploratory but prospective study like the present one has to be conducted to secure safety and test potential efficacy of an unapproved medical device on a rare intractable disease like LHON, particularly with an mtDNA 11778 mutation, because of the scarcity of the number of devices and participants for enrollment. At the time of conceptualization of this study, we judged it infeasible to design a confirmative prospective study like an RCT with a placebo control.
We repeat that this study is an exploratory pilot study, which serves as a base to promote a future confirmative RCT, and that we do not rigorously conclude the efficacy of SES for the treatment of LHON from the results of this study. We have, therefore, revised the manuscript title to convey the same.
In contrast, the protocol of this study has been approved by the Ethical Committee that is authorized under the Japanese Government Clinical Trail Act of April 2018 and has been disclosed in public on two registry sites (The University Hospital Medical Information Network-Clinical Trial Registry with a registration identifier of UMIN000031057 and The Japan Registry of Clinical Trials with a registration identifier of jRCTs052180066) as described in the Materials and Methods section. Therefore, adding a new control arm at this point trespasses this approved and opened protocol.
In addition, random allocation of participants to a control group inevitably requires resetting of a treatment group, resulting in discarding the present data and keeping those undisclosed. Whether the result is positive or negative, intentional closure of completely conducted data discards ethical responsibility to study participants and infringes on the ethical conduct of the clinical study.
Even though exploratory, studies on gene therapy (Feuer WJ et al., Gene Therapy for Leber Hereditary Optic Neuropathy: Initial Results. Ophthalmology, 123, 558-570, 2016; Guy et al., Gene Therapy for Leber Hereditary Optic Neuropathy: Low- and Medium-Dose Visual Results. Ophthalmology, 124, 1621-1634, 2017) and EPI administration (Sadun AA et al., Effect of EPI-743 on the clinical course of the mitochondrial disease Leber hereditary optic neuropathy. Arch Neurol 69, 331-338, 2012) have been published in high impact journals like the Journal of Clinical Medicine. We believe that this is because these studies have a potentially high clinical significance despite the exploratory nature and small sample size, considering the rarity and intractability of LHON. In this regard, we believe that the present study with minimal invasiveness has a sufficiently potential impact on LHON treatment, which is not only a basis for a future RCT but also suitable and within the scope of the prestigious Journal of Clinical Medicine.
As pointed out, participants who have no prior experience of SES might be controls unless the information of phosphene is given beforehand. In addition, sham stimulation, i.e.; 0 mA, might be applied to participants allocated to a so-called “control group” (Shatz A et al., 2017, Naycheva L et al., 2013). Strictly speaking, however, these groups may not be true controls because participants allocated to a treatment group have to be informed regarding the phosphene phenomenon to set a threshold stimulus intensity, and they not only notice phosphene but also feel skin irritation during electrical stimulation. In contrast, participants accolated to the control group feel nothing at all during stimulation. The point is that researchers who conduct the treatment know whether the particular participant is allocated to the control or the treatment group. Such an asymmetrical approach may likely let participants notice which group they are allocated during a series of treatment sessions.
Alternatively, we are now preparing for another exploratory study in which the frequency and period of stimulation session are increased to test whether a stronger and prolonged effect of visual recovery can be obtained by SES treatment. If we obtain stronger evidence form this approach, we will plan a multicenter RCT that recruits more number of participants and is designed as a confirmative study.
We have revised the Discussion section to reflect the above issues.
Minor comments:
Line 43: as mentioned previously, TES resulted in no improvement in visual function in patients with RP, and did not improve visual function, only a wave slopes in RAO.
Response: According to your suggestion, we have deleted the sentence describing the visual improvement of RP with TES.
Figure 2: The individual traces will need to be slightly offset, as several traces are not visible at all
Response: We have redrawn Figure 2a as per your suggestion.
Line 229: Please correct the numbering of cases with improved visual fields (case 5, 7, and 10)
Response: Thank you for pointing this out. We have corrected the number and changed the order of the grayscales in Figure 4c
Line 402-406: As stated above, I do not agree that a sham procedure is impossible. More cases and longer follow up do not eliminate a placebo effect. You can state that you, as a group of authors, did not wish to include a placebo arm, but should not state that it is not possible.
Response: According to your suggestion, we have added a sentence describing our plan to perform a comparative study in order to deny the placebo effects as the next clinical investigation (lines 394 and 396).
Line 412: Please state that the deterioration in vision in case 9 could have been due simply to natural disease progression.
Response: As you rightly pointed out, the deterioration of visual function may have been caused by the natural disease progression, although 8 months had passed without visual improvement after the onset, which fulfilled inclusion criteria. Thus, we have modified the sentence to state this possibility (lines 400–403 and 421).
